# Archaeal Lipids Regulating the Trimeric Structure Dynamics of Bacteriorhodopsin for Efficient Proton Release and Uptake

**DOI:** 10.3390/ijms23136913

**Published:** 2022-06-21

**Authors:** Sijin Chen, Xiaoyan Ding, Chao Sun, Fei Wang, Xiao He, Anthony Watts, Xin Zhao

**Affiliations:** 1Department of Physics, School of Physics and Electronic Science, East China Normal University, 500 Dongchuan Road, Minhang District, Shanghai 200241, China; 52194700015@stu.ecnu.edu.cn (S.C.); xiaoyan.ding@bioch.ox.ac.uk (X.D.); csun@phy.ecnu.edu.cn (C.S.); 51194700053@stu.ecnu.edu.cn (F.W.); 2Department of Biochemistry, University of Oxford, South Park Road, Oxford OX1 3QU, UK; 3Shanghai Engineering Research Center of Molecular Therapeutics and New Drug Development, Shanghai Frontiers Science Center of Molecule Intelligent Syntheses, School of Chemistry and Molecular Engineering, East China Normal University, Shanghai 200062, China; 4School of Chemistry and Molecular Engineering, NYU-ECNU Center for Computational Chemistry at NYU Shanghai, East China Normal University, Shanghai 200062, China

**Keywords:** bacteriorhodopsin, archaeal lipids, S-TGA-1 and PGP-Me, lipid–protein interactions, trimer stability, proton release and uptake

## Abstract

S-TGA-1 and PGP-Me are native archaeal lipids associated with the bacteriorhodopsin (bR) trimer and contribute to protein stabilization and native dynamics for proton transfer. However, little is known about the underlying molecular mechanism of how these lipids regulate bR trimerization and efficient photocycling. Here, we explored the specific binding of S-TGA-1 and PGP-Me with the bR trimer and elucidated how specific interactions modulate the bR trimeric structure and proton release and uptake using long-term atomistic molecular dynamic simulations. Our results showed that S-TGA-1 and PGP-Me are essential for stabilizing the bR trimer and maintaining the coherent conformational dynamics necessary for proton transfer. The specific binding of S-TGA-1 with W80 and K129 regulates proton release on the extracellular surface by forming a “Glu-shared” model. The interaction of PGP-Me with K40 ensures proton uptake by accommodating the conformation of the helices to recruit enough water molecules on the cytoplasmic side. The present study results could fill in the theoretical gaps of studies on the functional role of archaeal lipids and could provide a reference for other membrane proteins containing similar archaeal lipids.

## 1. Introduction

Biomembranes consist of lipids, proteins, and carbohydrates and are fundamental to life. They are responsible for the cell envelopes that enable cells to absorb nutrients and exclude the most harmful agents from entering cells. They are also very dynamic in terms of structure, enabling interactions among proteins and between proteins and lipids to provide temporal associations that are significant to membrane functions [1]. The basic unit of a membrane is a bilayer that is formed by phospholipids and sphingolipids that are organized into two layers with their polar headgroups along the two surfaces and their acyl chains creating a nonpolar domain in between.

Transmembrane proteins are embedded in the lipid bilayer and play a crucial role in biological processes such as ion transport, signal transduction, and communication, making ideal drug targets [1,2]. Phospholipids are indispensable to transmembrane proteins, and increasing evidence indicates that the functions of transmembrane proteins are affected by different lipid environments [3,4,5,6,7,8] and specific lipid–protein interactions [9,10,11,12,13,14,15]. However, only a few high-resolution crystallographic data describing the details of protein–lipid interactions in transmembrane proteins are available. For example, the lipid polar headgroups in the K^+^ channel protein Kv2.1 [16] and light-adapted bacteriorhodopsin (bR) have not yet been well resolved [17,18]. In addition, not many quantitative methods are available to identify specific lipid–protein interactions. 

bR is a typical seven-transmembrane (7TM) light-driven proton pump found in the *Halobacterium halobium* (*H. halobium*) [19] and has served as an ideal model for studying the ion transport and energy conduction in membrane proteins for many years [20,21,22,23,24]. The retinal chromophore is covalently bound to a lysine residue on helix G and forms a protonated Schiff base (SB) linkage. The photoisomerization of all-trans retinal to the 13-cis isomer triggers conformation changes in the receptor to release a proton to the extracellular surface and uptake another one from the cytoplasmic bulk through a series of photo-intermediate states, including the intermediate states of K, L, M_1_, M_2_, M_2_’, N, and O [24,25,26,27]. A detailed description of the bR photocycle and the proton transfer steps is shown in Appendix A Appendix A. 

The bR purple membrane (PM) consists of approximately 90% polar lipids: 60% phospholipids and 30% glycolipids; the remaining 10% are mainly neutral squalene lipids [28,29,30,31]. Phospholipids are primarily made of phosphatidylglycerol (PG), phosphatidylglycerol sulfate (PGS), and phosphatidylglycerol phosphate methyl ester (PGP-Me). Glycolipids are mainly composed of archaeal glycocardiolipin (GlyC), cardiolipin (bisphosphatidylglycerol, BPG), and glycolipid sulfate (3-HSO_3_-Galp-β1,6-Manp-α1,2-Glcp-α1,1-sn-2,3-diphytanylglycerol, S-TGA-1). S-TGA-1 is a specific lipid located in the center of the bR trimer closure and is essential for intra-trimer stabilization [32,33,34,35]. PGP-Me is a particular lipid existing in the space of the adjacent bR monomers and acts as a glue to bridge contact between the bR monomers and maintain trimer stabilization [34,35,36,37,38] (Figure 1). Previous studies have revealed that these two polar lipids are essential to bR trimerization and proton transfer through specific interactions [17,32,33,34,35,37,38,39,40,41,42,43,44,45,46,47,48,49,50,51,52,53,54,55]. However, the detailed coupling mechanism of these specific lipids with the bR proton pumping function during proton release and uptake remains unclear due to the lipids being inaccessible commercially or for laboratory synthesis.

On the other hand, molecular dynamic (MD) simulations are an ideal tool for bridging the gap by exploring specific lipid–protein interactions [56,57,58,59,60,61,62,63]. However, simulations in the early years were often confined to a single or a few mixed lipids [14,64,65], and the simulation time was not long enough (1–100 ns) [57,59,66]. In this paper, we carry out a long-term all-atom MD simulation (1 μs) with the single-point mutagenesis to delineate the molecular mechanism of the specific interaction of S-TGA-1 and PGP-Me with bR during the proton release and uptake processes. Amber18 [67] and Gaussian16 [68] are used to construct the S-TGA-1 and PGP-Me force fields, which are then embedded into the corresponding position of the protein via the PyMOL [69] and PPM web servers [70]. Our results provide an improved understanding of how archaeal lipids influence the bR trimeric structure and proton pumping function and provide inspiration for studying lipid–protein interactions in other membrane proteins, including these archaeal lipids. Appendix A Appendix A summarizes the 10 simulation models used. 

## 2. Results and Discussion

The dynamic functions of any membrane protein directly correspond to the conformational dynamics in the native membrane environment. Many studies have reported that the photocycle kinetics of bR are different in the native membrane and in the membrane mimics [18,38,44,51,52]. S-TGA-1 and PGP-Me are the specific archaeal lipids of PM, playing an important role in this difference by binding the bR. However, their interactions with the bR trimer have not been well resolved by X-ray crystallography [18], and so far, no long-term MD simulations (microsecond time scale) are available to gain insight into the dynamic binding of S-TGA-1 and PGP-Me with the bR trimer. Here, we sought to determine the specific interactions of S-TGA-1 and PGP-Me with the bR trimer through all-atom MD simulations over a length of 1 μs.

### 2.1. Maintaining the Overall Coherent Dynamics by S-TGA-1 and PGP-Me Binding

The bR trimer achieved a smaller root mean square deviation (RMSD) with the lipids of POPC, S-TGA-1, and PGP-Me (denoted here as M1) than it did with only POPC lipids (M2) throughout the simulations (Figure 2A). The overall root mean square fluctuation (RMSF) of M1 was also less than M2 (Figure 2D and Appendix A). All of the results indicate that S-TGA-1 and PGP-Me are conducive to bR trimer stabilization with appropriate dynamics. Previous research also mentioned the importance of S-TGA-1 and PGP-Me for the bR 2D crystal alignment and trimeric stabilization [34,36,42,44,52]. The effects of these two archaeal lipids on the dynamics of the bR trimer were then further examined separately by the membranes containing only the bR trimer, POPC, and S-TGA-1 (M3) or the bR trimer, POPC, and PGP-Me (M4). Our results showed that both S-TGA-1 and PGP-Me could stabilize the trimer’s dynamic conformation, with S-TGA-1 being more prominent (Figure 2B,C,E,F), a finding that was consistent with the previous results [52]. In addition, the simulations showed that S-TGA-1 contributes to the order of the B-C loop, D-E loop, F-G loop, and helix D; PGP-Me promotes the order of the C-D loop, E-F loop, and helix E (Figure 2E,F and Appendix A). As such, we suggest that S-TGA-1 and PGP-Me enhance the bR trimer’s stability by mediating the dynamics of the extracellular and cytoplasmic sides, respectively. Several studies have confirmed that S-TGA-1 and PGP-Me are essential for the bR’s array structure and stabilization [34,36,44,52].

A more uniform and coherent correlation in M1, but an incoherent and localized correlation in M2, was observed by the dynamic cross-correlation map (DCCM) and dynamic cross-correlation network (DCCN) analyses [71,72,73,74] (Figure 2G,J,K). The differences observed in M2 were also identified in M3 and M4; although the overall correlation was not as pronounced as in M1 (Figure 2H,I,L,M). Repeated simulations also observed similar results, as shown in Appendix A. These results indicate that S-TGA-1 and PGP-Me are conducive to stabilizing the bR trimer and have more consistent dynamics for different functional domains. Previous studies have also suggested that archaeal lipids are needed for the adjoining monomers to interact and form a trimer [38,41]. The absence of S-TGA-1 disturbed the dynamic correlation of Helix D and the cytoplasmic region (E-F loop), and the lack of PGP-Me affected the dynamic correlation of the extracellular region of Helix D, indicating that the regulation of the bR dynamic conformation by S-TGA-1 and PGP-Me was synergistic instead of independent. These observations suggested that S-TGA-1 and PGP-Me can maintain the appropriate dynamics to ensure the bR trimer has enough coherent flexibility for proton release and uptake on both sides of the surfaces [18,38,44,51,52].

The headgroup of S-TGA-1 interacted with Y64, L66, T67, W80, and K129 through hydrogen bond (H-bond) interactions; the indole ring of W80 was almost parallel to the trimer axis due to its Pi–alkyl hydrophobic interaction with the 2-phytanol tail of S-TGA-1; G113, G116, and G120 formed a groove to accommodate the 3-phytanol tail through hydrophobic interactions on the extracellular side (Figure 3A,B). The lifetime of these H-bonds indicated that the contact remained intact throughout all of the simulations, with the exception of a slightly weaker H-bond between T67 and the 3-hydroxy of mannose (Figure 3C). K129 may form a salt bridge with the galactosyl-3-sulfate group of S-TGA-1 (Figure 3D). Some of the above interactions were also mentioned in earlier crystal structure studies [17,32,75]. In addition, we identified extra hydrophobic interactions involving V69, P70, A84, L87, F88, and V124 that have not been previously reported in any crystal structures (Figure 3A). On the cytoplasmic side, the PGP-Me headgroup had H-bond interactions with K40, G155, and T157 (Figure 4A,B), with a lifetime of about 1 μs for G155 and T157 and a lifetime longer than 700 ns for K40 (Figure 4C); and the ε-amino group of K40 may come into contact with the polar headgroup of PGP-Me via a salt bridge (Figure 4D). In addition, several residues on the cytoplasmic side of helices B, D, and E (L28, M32, Y43, A44, T47, A51, A110, A114, I117, I140, A143, A144, Y147, and Y150) had hydrophobic contact with both of the phytanol tails of PGP-Me (Figure 4A). These specific lipid–protein interactions provide connections between the adjacent monomers on both ends of the bR trimer to maintain the global conformational dynamic stability of the whole protein complex for proton release and uptake via coherent correlations. 

### 2.2. Coupling of S-TGA-1 with the Extracellular Surface of bR for Proton Release

Previous studies have reported that S-TGA-1 and PGP-Me are essential for bR to carry out its normal photocycle and proton pumping activity [38,39,42,44,49,51,52,53]. The results above also indicated that the archaeal lipids could regulate the stability and dynamic conformation of the bR trimer. However, it remains unclear how S-TGA-1 and PGP-Me interact with the bR trimer to regulate its function-related dynamic conformation. Here, we further explored how specific lipid–protein interactions influence bR proton release and uptake according to a series of simulations on different mutants.

Further analysis of M1 revealed that the interactions between E194 and E204, the two crucial residues for proton release, were perturbed by S-TGA-1. In the presence of S-TGA-1, our simulations showed that E194 and E204 are close enough to form a “Glu-shared” model to cast the proton between them for effective proton release (Figure 5A,B and Appendix A), as reported previously [76,77,78,79,80]. Previous studies showed that A51 is stationary during simulations [81], so the distance from A51 to R82 can be used to monitor R82 movement during the photocycle. It is clear that R82 couples with the retinal binding pocket in dark-adapted bR and affects the proton release complex (PRC) for proton release in the M state [75]. Thus, we may be able to indirectly investigate the influence of S-TGA-1 on E194 and E204 through the distance changes from Y83 to E194 and from A51 to R82. Interestingly, the distances from Y83 to E194 and from A51 to R82 were not much different in M1 and M2 (Figure 5C,D and Appendix A), but there were apparent displacements of both W80 and Y83 on helix C towards E204 (Figure 5A), implying that the displacements of W80 and Y83 have an impact on E194∙∙∙E204 packing. Such shifts may originate from W80’s pi–alkyl hydrophobic interaction with S-TGA-1. In addition, R134 has H-bond interactions with T128 on the D-E loop, E194 on the F-G loop, and K129 on the D-E loop to produce cascade perturbations on the F-G loop [75,82]. Therefore, it is worth considering whether the interaction of K129 with S-TGA-1 also affects the “Glu-shared” model. To elucidate the underlying molecular mechanism of how W80∙∙∙S-TGA-1 and K129∙∙∙S-TGA-1 specific interactions regulate the proton release, we constructed simulations on W80A (M5, the membrane containing the W80A-bR trimer, POPC, S-TGA-1, and PGP-Me lipids) and K129L (M6, the membrane containing the K129L-bR trimer, POPC, S-TGA-1, and PGP-Me lipids) mutations. 

The RMSDs of M5 and M6 over the total simulation were more significant than that of M1, and the same for RMSFs of the C-D loop, helices F and G, with the difference caused by W80A more prominent (Figure 6A–D). These results indicate that W80 and K129 could mediate the connection between the adjacent monomeric helices by binding with S-TGA-1 to maintain appropriate dynamics for the bR trimer. DCCM and DCCN analyses showed that the change in inter-residue interactions caused the negative correlations to increase significantly, especially on the extracellular surface (Figure 6E–H), resulting in reduced stability for the bR trimer. 

We further monitored the detailed changes of several key residues and water molecules on the extracellular side caused by the W80A and K129L mutations. For the membrane of the W80A-bR trimer with S-TGA-1 and PGP-Me (M5), E194–E204’s “Glu-shared” pattern was not observed (Figure 7A and Appendix A), Y83 still formed an H-bond with E194, and R82’s guanidine group demonstrated little change (Figure 7B,C and Appendix A). However, Y83 underwent a significant displacement, moving to a position similar to what we observed in M2 (Figure 7D,E), indicating that the hydrophobic interactions between W80 and S-TGA-1 may regulate Y83 conformation to encourage E194 to be closer to E204. The H-bond between E194 and E204 fluctuated in M7 (membrane of the W80A-bR trimer without S-TGA-1 and PGP-Me), and the Y83 and R82 displacements were also significantly different from what we observed in M5 (Appendix A, Appendix A). All further evidence suggested that the Pi-Alkyl hydrophobic interaction between W80 and S-TGA-1 promotes the formation of the “Glu-shared” model favorable for proton release through regulating Y83. Weik et al. have proposed that the interactions between aromatic residues, especially tryptophan, and lipids are essential for bR’s proton pumping function [49]. Our present simulation results more concretely illustrate this proposal.

The H-bond interaction between E194 and E204 was also perturbed in the M7 of the K129L-bR trimer with the lipids of POPC, S-TGA-1, and PGP-Me (Figure 8A and Appendix A); however, this perturbation was not observed in the M8 of the K129L-bR trimer without S-TGA-1 and PGP-Me (Appendix A and Appendix A), suggesting the changes of the “Glu-shared” model were due to the loss of contact between K129 and S-TGA-1. The H-bond between R134 and E194 was unstable (Figure 8B and Appendix A), and there were more water molecules around K129 and T128 in M6 (Figure 8C), but this was not the case in M8 (Appendix A, and Appendix A). Therefore, we proposed that the broken H-bond interaction between K129 and S-TGA-1 could attract more water molecules to gather around K129 and T128, causing increased fluctuations in the D-E loop. The F-G loop was perturbed through agitation in the H-bond chain of T128∙∙∙R134∙∙∙E194; thereby destabilizing the H-bond interaction between E194 and E204 (Figure 8D,E). The significantly increased fluctuation and incoherent correlations of the D-E and F-G loops agree further with our proposal.

It is well known that proton release occurs in the M state, and the rapid release is beneficial for accelerating the M state decay and improving the proton pumping efficiency [24,83,84]. Our simulations demonstrated that W80 and K129 significantly influence E194∙∙∙E204 to form the “Glu-shared” model through interacting with S-TGA-1, which is conducive to promoting proton release during the short lifetime of the M state [49,52] (Appendix A).

### 2.3. Coupling of PGP-Me with the Cytoplasmic Surface of bR for Proton Uptake 

Formation of the transient water chain and having an “open” conformation in the cytoplasmic half channel are essential for the proton uptake in the later stage of the bR photocycle [85,86]. Any changes in the opening and closing of the helix conformation and the number of water molecules induced by PGP-Me may affect the proton uptake. On the cytoplasmic side, the helices A, B, D, E, and the E-F loop in M1 were quite different from M2 according to our simulations (Figure 9A), but were consistent with the bR crystal structure containing PGP-Me (Appendix A) [32,35]. A further comparison found that PGP-Me resulted in a larger opening (Figure 9B and Appendix A, and Appendix A), and such conformational differences significantly impacted the number of water molecules in the accessible region of the cytoplasm. More water molecules aggregated around the hydrophobic residues on the cytoplasmic surface of M1 than M2 (Figure 9C,D). 

Otomo et al. found that bR cannot self-assemble in vivo to form PM upon the mutation of K40 to a neutral residue, resulting in an impaired proton pumping function [87]. Therefore, we performed simulations on the K40L-bR trimer (M9, the membrane containing the K40L-bR trimer, POPC, S-TGA-1, and PGP-Me lipids) to explore how the K40∙∙∙PGP-Me interaction affects the protein trimer stability and proton uptake. The RMSD of the K40L-bR trimer was slightly higher than that of M1, implying that the specific K40∙∙∙PGP-Me interaction may contribute to stabilizing the bR trimer (Figure 10A). The number of water molecules on the cytoplasmic surface increased, with helix A moving away from its initial position and having a more flexible open conformation (Figure 10B,C and Appendix A). However, a slight change was observed in the K40L mutant without S-TGA-1 and PGP-Me in M10 (Appendix A, Appendix A). Therefore, we speculate that the interaction between K40 and PGP-Me may cause changes in the helices and the number of water molecules at the cytoplasmic terminal. In the late M state of bR, the cytoplasmic side of helices E and F slope outward to open a channel for water molecules [85,88,89,90]. However, our results showed that the K40L mutant had little effect on the cytoplasmic side of helices B, D, and E (Figure 10D–F and Appendix A), suggesting the main consequence of the mutant is to make sufficient preparations for the water chain required for the proton uptake. Our flashlight-induced transient absorption spectroscopy study on the native membrane of K40L-bR supported this proposal, showing shortened M-state decay and a shorter ground state recovery time (Appendix A). Additionally, it is worth noting that the apparent decrease in the intensity of the M state indicates that the proton pumping capacity of bR was weakened by the K40L mutant (Appendix A), which could be due to the more negative correlation caused by the K40L mutation (Appendix A). 

## 3. Conclusions

For the first time, we provided detailed structural information about the specific interactions of S-TGA-1 and PGP-Me with bR. We demonstrated that these interactions promote the stability of the bR trimer and regulate proton transport by ensuring coherent conformation dynamics of bR and elucidated the underlying molecular mechanism by which key lipid–protein interactions affect proton uptake and release. On the extracellular side, W80 and K129 significantly influenced E194∙∙∙E204 packing to form a “Glu-shared” model through interactions with S-TGA-1, making it conducive to facilitating proton release (Figure 11A). Removing those specific interactions in W80A and K129L disrupted the formation and decay of the photo-intermediate states, especially proton release, and further impeded the recovery of the ground state and the desensitization time (Appendix A). On the cytoplasmic side, the interaction between K40 and PGP-Me mainly affected bR photocycling by regulating the number of water molecules. The K40L mutation increased the number of water molecules around the hydrophobic residues, promoting faster proton uptake and shortening the photocycling lifetime (Figure 11B), which is the desired property for optogenetic tools [91,92].

This work demonstrates that the long-term and all-atom molecular dynamics simulation is a powerful method to connect the overall structural stability and conformational dynamics of a large membrane protein complex with the dynamic function of the protein. The strategy used here may be broadly applied to study the dynamic regulation of the function-related conformation changes induced by the specific binding of other membrane receptors, such as lipids or ligand bindings.

## 4. Materials and Methods

### 4.1. Molecular Models and Parameters

The light-adapted state X-ray structure of bR with an all-*trans* retinal chromophore (PDB ID: 1C3W [75], with internal water) was used as a template in our calculations and simulations. For mutant simulations, we mutated the corresponding residues by PyMOL [69]. The CHARMM-GUI [93,94,95,96] server was used to build the initial structure of the simulations, including the explicitly hydrated bilayer, water molecules, and neutralized ions. As reported previously, the POPC (1-palmitoyl-2-oleoyl-sn-glycero-3-phosphocholine) lipid bilayer could reasonably maintain the interactions between residues in the bR trimer and the water dynamics [65,86,97,98,99] and its topology and parameters were tested extensively [100]. Therefore, all of our simulations (M1-M10) were performed in a POPC system constructed with 156 lipids (78 in the upper leaflet and 77 in the lower brochure) and about 10,000 water molecules. S-TGA-1 and PGP-Me were added to the corresponding positions in the bR trimer using TM-align [101] and PyMOL based on the information of GLC-1, MAN-2, SGA-3, ARC-1002, and GOL-1003 in 1BRR [32], and 2DP-267 in 2AT9 [35]. Each simulation system contained a bR trimer and 3 archaeal phospholipids S-TGA-1 and PGP-Me units. The missing residues in the loop regions of the crystal structure were built using the Schrödinger program [102], and the trimeric forms were generated from the symmetrical information of the crystal structures using VMD [103]. The position of the simulation model in the lipid membrane was determined by the PPM web server [70]. Asp96, Asp115, Glu204, and the Schiff base were protonated. 

The retinal was treated as a single unit, and the parameters were obtained using the antechamber module and the general force field (GAFF) [104] in the AMBER program. Additionally, the parameters for S-TGA-1 and PGP-Me were obtained by combining Amber18 [67] and Gaussian16 [68]: First, we performed geometric optimization on S-TGA-1 and PGP-Me using the B3LYP/6-31G** level, and we then calculated the optimized structural singlet point energy, extracted the RESP2 charge [105], and fit the RESP2 charge and generated a small molecule force field file using the antechamber module in Amber18 [67]. The POPC lipid molecules were assigned to the Lipid14 force field [106]. The parameters of protein residues were assigned based on the AMBER ff14SB protein [107], and the TIP3P model [108] was utilized for the water molecules.

The final simulated systems were constructed using the XLEAP program [109]. MD simulations were carried out using the pmemd.cuda module in the AMBER18 program [67]. MD simulations with periodic boundary conditions were conducted in the NPT ensemble at 303 K and at 1 bar. Long-range electrostatic interactions were treated using the PME method [110], and empirical Lennard-Jones potentials were used to calculate the short-range van der Waals interactions with a cutoff of 10 Å. The temperature was regulated by Langevin dynamics [111] with a collision frequency of 1.0 ps^−1^, the anisotropic Berendsen weak-coupling method [112] was utilized to couple the system to a barostat of 1 bar, and the SHAKE algorithm [113] was used to deal with the vibrations involving hydrogen atoms.

### 4.2. Molecular Dynamics Simulation

MD simulations were carried out using the pmemd.cuda module in the AMBER18 program [67]. First, we performed the simulations with everything fixed except the lipid tails to induce the appropriate disorder of a fluid-like bilayer. Then, we performed energy minimization and the equilibration of the bR-POPC (including bR-POPC-mutants) and bR-S-TGA-1_PGP-Me (including bR-S-TGA-1_PGP-Me-mutants) in two phases. Finally, the long-term unrestrained MD simulations continued when all of the atoms were free, and the coordinates were saved every 4 ps. The energy minimization and equilibration processes are described in detail below.

For the bR-S-TGA-1_PGP-Me and bR-S-TGA-1_PGP-Me-mutant simulations, each of the systems needed three-stage energy minimization. In the first stage, the protein, crystal water, and lipid bilayer (POPC) were restrained by a harmonic potential with a force constant of 10.0 kcal/mol·Å^2^, S-TGA-1 and PGP-Me were controlled by a force constant of 30.0 kcal/mol·Å^2^, and all of the other atoms were unrestrained; in the second stage, the protein, crystal water, and lipid bilayer (POPC) were restrained by a harmonic potential with a force constant of 5.0 kcal/mol·Å^2^, S-TGA-1 and PGP-Me were controlled by a force constant of 20.0 kcal/mol·Å^2^, and all of the other atoms were unrestrained; in the third stage, S-TGA-1 and PGP-Me were also restrained by a harmonic potential with a force constant of 20 kcal/ mol·Å^2^, and the remaining atoms were relaxed. Energy minimization was executed for 100,000 steps per stage, and the steepest descent method was used for the first 60,000 steps, which was then switched to the conjugate gradient method for the rest of the steps. After energy minimization, the whole system carried out three stages of equilibration. In the first stage, the systems were gradually heated from 0 to 100 K at 400 ps in the NVT ensemble, and the bR protein and crystal water were restrained by a harmonic potential with force constant of 10.0 kcal/mol·Å^2^, S-TGA-1 and PGP-Me were restrained by a harmonic potential with a force constant of 100.0 kcal/mol·Å^2^, and the lipid bilayer (POPC) was controlled by a force constant of 5.0 kcal/mol·Å^2^; in the next stage, the systems were gradually heated from 100 to 200 K at 400 ps in the NVT ensemble, and the bR protein and crystal water were restrained by a harmonic potential with a force constant of 8.0 kcal/mol·Å^2^, S-TGA-1 and PGP-Me were restrained by a harmonic potential with force constant of 300.0 kcal/mol·Å^2^, and the lipid bilayer (POPC) was controlled by a force constant of 2.5 kcal/mol·Å^2^; in the third stage, the systems were gradually heated from 200 to 303 K at 400 ps in the NVT ensemble, and the restraints of force constant on the bR protein and crystal waters reduced to 4.0 kcal/mol·Å^2^, the force constant on the lipids (POPC) reduced to 2.0 kcal/mol·Å^2^, and S-TGA-1 and PGP-Me were restrained by a harmonic potential with a force constant of 500.0 kcal/mol·Å^2^. Then, we performed three 2 ns equilibrations with the NPT ensemble, where the constant pressure was 1.0 bar and where there were progressively decreasing force constant constraints on S-TGA-1 and PGP-Me, which, from first to last, were 300.0 kcal/mol·Å^2^, 100.0 kcal/mol·Å^2^, and 20.0 kcal/mol·Å^2^. 

For the bR-POPC and bR-POPC-mutant simulations, each system needed two-stage energy minimization. In the first stage, the protein, crystal water, and POPC were restrained by a harmonic potential with a force constant of 10.0 kcal/mol·Å^2^, while all of the other atoms were unrestrained; in the second stage, all of the atoms were relaxed. Energy minimization was executed for 100,000 steps per stage, and the steepest descent method was used for the first 50,000 steps, which was then switched to the conjugate gradient method for the rest of the steps. After energy minimization, the whole system carried out three equilibration stages. In the first stage, the systems were gradually heated from 0 to 100 K at 500 ps in the NVT ensemble, in which the bR protein and crystal water were restrained by a harmonic potential with a force constant of 5.0 kcal/mol·Å^2^, and the POPC was controlled by a force constant of 2.5 kcal/mol·Å^2^; in the next stage, the systems were gradually heated from 100 K to 303 K at 500 ps in the NVT ensemble, and the bR protein and crystal water were restrained by a harmonic potential with a force constant of 2.5 kcal/mol·Å^2^, and the POPC was controlled by a force constant of 1.0 kcal/mol·Å^2^; in the third stage, a 1 ns equilibration was performed at the NPT ensemble and at a constant temperature of 303 K, and the bR protein and crystal water were restrained by a harmonic potential with a force constant of 1.0 kcal/mol·Å^2^, and the POPC was controlled by a force constant of 0.5 kcal/mol·Å^2^. Then, we performed 10 ns unrestrained equilibrations with the NPT ensemble with the continuous pressure set at 1.0 bar.

### 4.3. Analysis of the Simulations

The cpptraj module of AmberTools18 was used to calculate the root mean square deviations (RMSD), root mean square fluctuation (RMSF), dynamic cross-correlation map (DCCM), the distances between key residues, the dihedral angle of the retinal, and the lifetime of the hydrogen bonds. The LigPlot+ diagrams [114] (including those for the hydrogen bonds and the van der Waals and hydrophobic interactions) were analyzed using HBPLUS [115] and LIGPLOT [116]. The dynamical cross-correlation networks were built based on the DCCM to represent the correlated motion between the residues in different domains. The Cα atoms of each residue interacted with each other in the form of a pair of nodes [117], and PyMOL was used to visualize the interaction network. All molecular graphic and video representations were created using PyMOL.

## Figures and Tables

**Figure 1 ijms-23-06913-f001:**
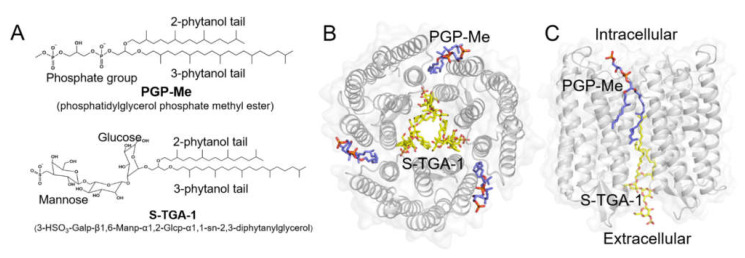
Schematic diagram of the structures of S-TGA-1, PGP-Me, and the bacteriorhodopsin (bR) trimer. (**A**) Molecular structures of S-TGA-1 and PGP-Me; (**B**,**C**) top and front view for the locations of S-TGA-1 (yellow sticks) and PGP-Me (dark blue sticks) within the bR trimer (grey cartoon).

**Figure 2 ijms-23-06913-f002:**
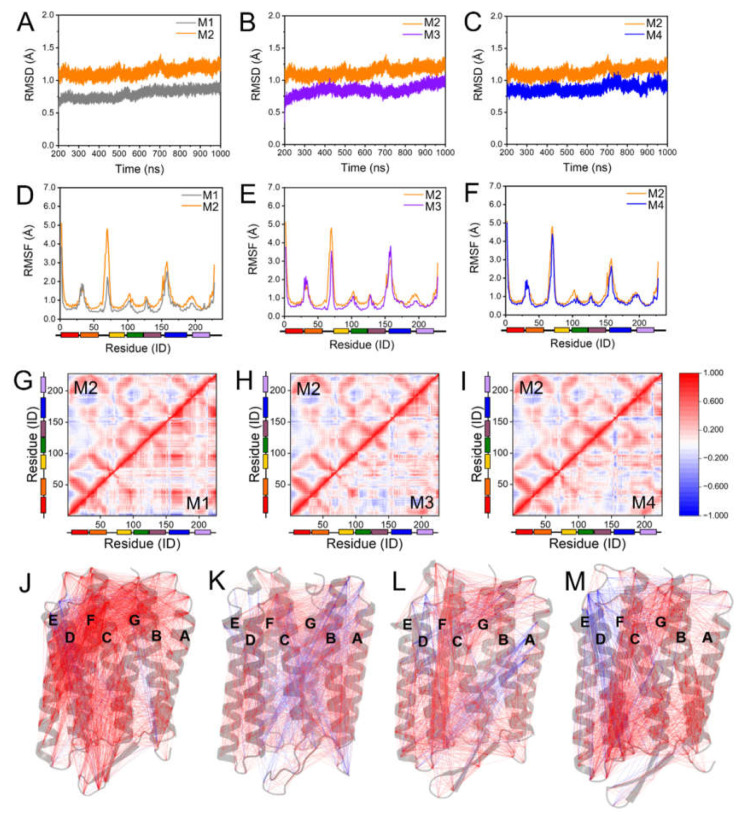
The influence of S-TGA-1 and PGP-Me on the dynamic stability of the bR trimer. (**A**–**C**) Time evolutions of the root mean square deviation (RMSD) of the Cα atoms during the simulations. (**A**) M1 (bR trimer with POPC, S-TGA-1, and PGP-Me, grey line) vs. M2 (bR trimer with POPC only, orange line); (**B**) M3 (bR trimer with POPC and S-TGA-1, purple line) vs. M2; (**C**) M4 (bR trimer with POPC and PGP-Me, blue line) vs. M2; (**D**–**F**) time evolutions of root mean square fluctuations (RMSF) of the Cα atoms: (**D**) M1 vs. M2; (**E**) M3 vs. M2; (**F**) M4 vs. M2; (**G–I**) dynamic cross-correlated map (DCCM) analyses of the Cα atoms; (**G**) M1 vs. M2; (**H**) M3 vs. M2; (**I**) M4 vs. M2. The upper left triangle represents M1. The color scale represents the correlation intensity from the maximum negative value (−1) to the ultimate positive value (+1); (**J**–**M**) dynamic cross-correlation network (DCCN) analyses of residue–residue cross-correlations from M1 to M4. The grey cartoon represents the bR structure of Monomer1 (one of the three monomers in the bR trimer), and blue and red lines indicate the negatively and positively correlated motions, respectively.

**Figure 3 ijms-23-06913-f003:**
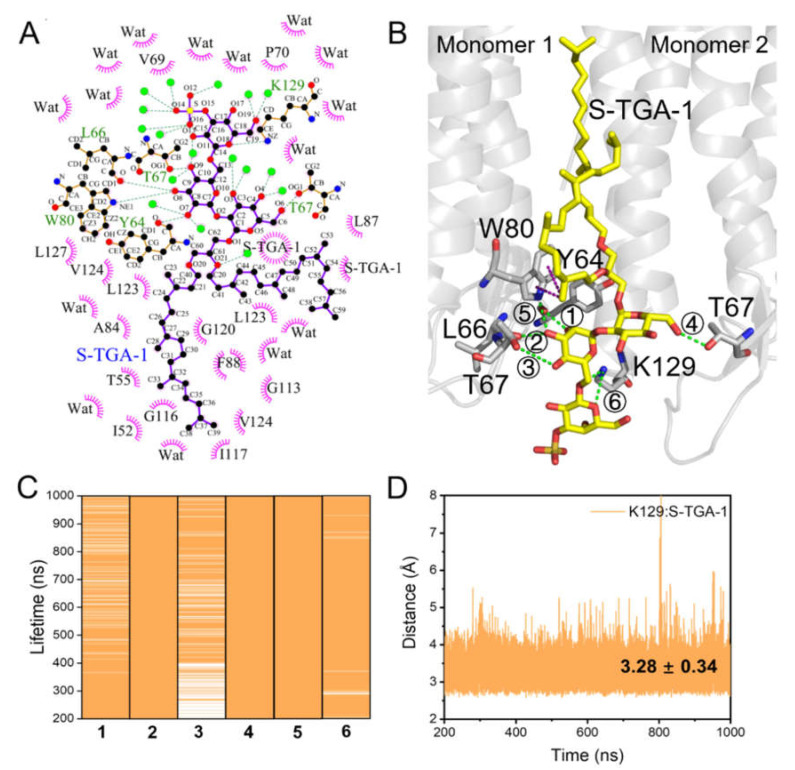
Schematic representation of S-TGA-1 binding to the bR trimer. (**A**) The LigPlot+ diagrams of the S-TGA-1 binding sites from the last snapshot of M1; the green dotted lines and the pink spoked arcs represent hydrogen bond (H-bond) and hydrophobic interactions, respectively; (**B**) the possible H-bond (green dotted line) and Pi–alkyl hydrophobic interactions (purple dotted line) between S-TGA-1 with bR. Monomer1 and Monomer2 represent the two adjacent monomers in the bR trimer. The grey sticks represent the interacting residues, and the yellow sticks represent S-TGA-1; (**C**) evaluation of the lifetimes of the H-bond interactions for S-TGA-1; see (**B**) for the description of the numbers: (1) Y64∙∙∙2-hydroxyl of mannose; (2) L66∙∙∙2-hydroxyl of mannose; (3) T67∙∙∙3-hydroxyl of mannose; (4) T67∙∙∙6-hydroxyl of glucose; (5) W80∙∙∙2-hydroxyl of mannose; (6) K129∙∙∙ether group of galactosyl-3-sulfate; (**D**) time evolution of the distance between K129 and the galactosyl-3-sulfate of S-TGA-1. These interaction modes were reproducible in the second time-repeating simulation, as shown in Appendix A.

**Figure 4 ijms-23-06913-f004:**
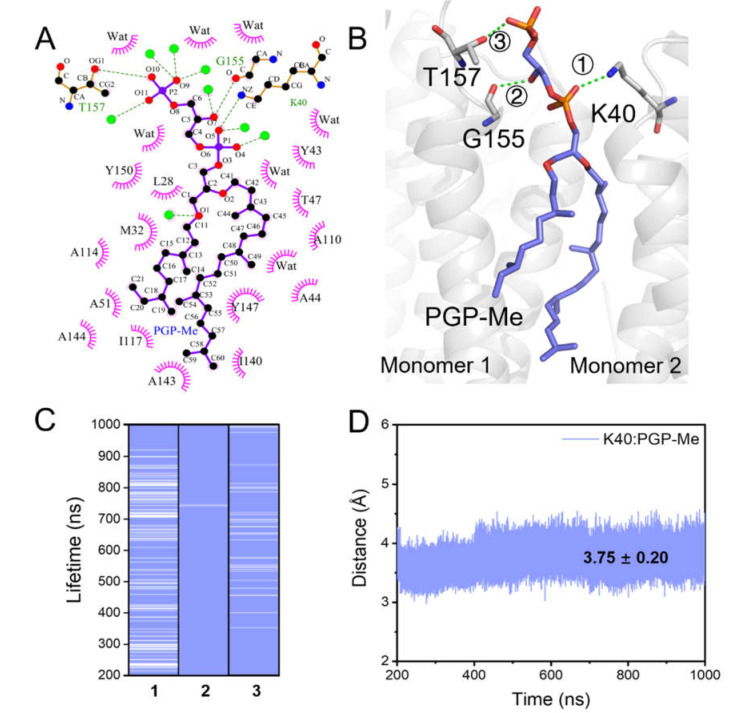
Schematic representation of PGP-Me binding to the bR trimer. (**A**) The LigPlot+ diagrams of the PGP-Me binding sites from the last snapshot of M1; the green dotted lines and the pink spoked arcs represent H-bond and hydrophobic interactions, respectively; (**B**) the possible H-bond interactions (green dotted line) between PGP-Me and the two adjacent monomers in the bR trimer. The grey sticks represent the residues interacting with PGP-Me via H-bond, and the dark-blue sticks represent PGP-Me; (**C**) evaluation of the lifetimes of the H-bond interactions for PGP-Me; see (**B**) for the description of the numbers: (1) K40∙∙∙O5 of the phosphate group; (2) G155∙∙∙O7 of the phosphate group; (3) T157∙∙∙O10 of the phosphate group; (**D**) the evolution of the distance between K129 and the galactosyl-3-sulfate of S-TGA-1. These interaction modes were reproducible in the second time-repeating simulation, as shown in Appendix A.

**Figure 5 ijms-23-06913-f005:**
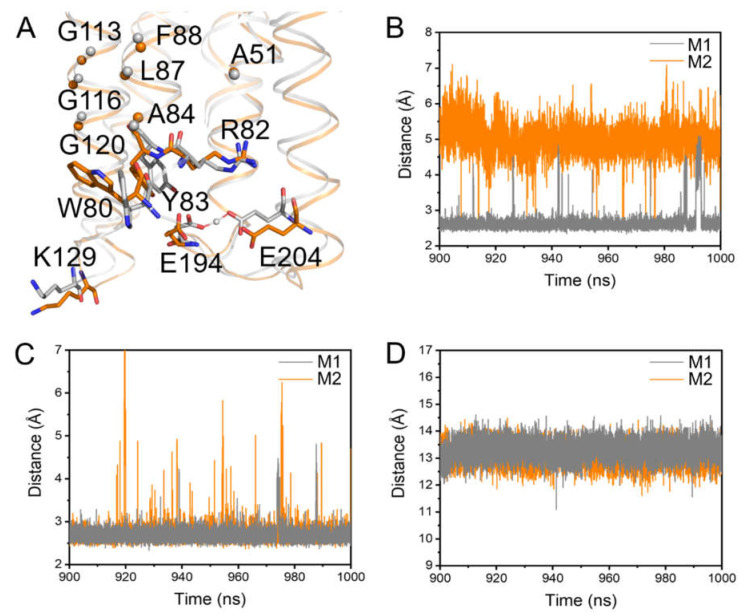
Conformation changes on the extracellular side induced by S-TGA-1. (**A**) Detailed changes in the key residues in the extracellular side in M1 (grey cartoon, sticks, and spheres) relative to M2 (orange cartoon, sticks, and spheres); (**B**–**D**) the evolution of the distance between the key residues from the last 100 ns simulations of M1 (grey line) and M2 (orange line): (**B**) E194-E204; (**C**) Y83-E194; (**D**) R82-A51. Repeated simulations could reproduce these behaviors (Appendix A and Appendix A).

**Figure 6 ijms-23-06913-f006:**
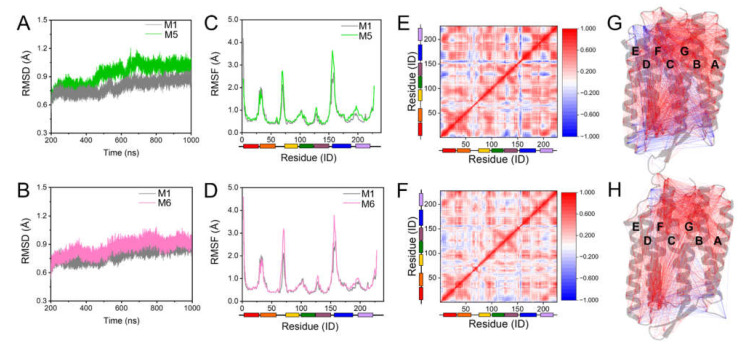
The influence of the W80∙∙∙S-TGA-1 and K129∙∙∙S-TGA-1 interactions on the dynamic stability of the bR trimer. (**A**,**B**) Time evolutions of the RMSDof the Cα atoms during the simulations; (**A**) M1 (grey line) vs. M5 (membrane containing the W80A-bR trimer, POPC, S-TGA-1, and PGP-Me lipids, green line); (**B**) M1 vs. M6 (membrane containing the K129L-bR trimer, POPC, S-TGA-1, and PGP-Me lipids, pink line); (**C**,**D**) time evolutions ofRMSF of the Cα atoms: (**C**) M1 vs. M5; (**D**) M1 vs. M6; (**E**,**F**) DCCM analyses of the Cα atoms for M5 and M6. The color scale represents the correlation intensity from the maximum negative value (−1) to the ultimate positive value (+1); (**G**,**H**) DCCN analyses of residue–residue cross-correlations for M5 and M6. The grey cartoon represents the bR structure of Monomer1, and blue and red lines indicate the negatively and positively correlated motions, respectively.

**Figure 7 ijms-23-06913-f007:**
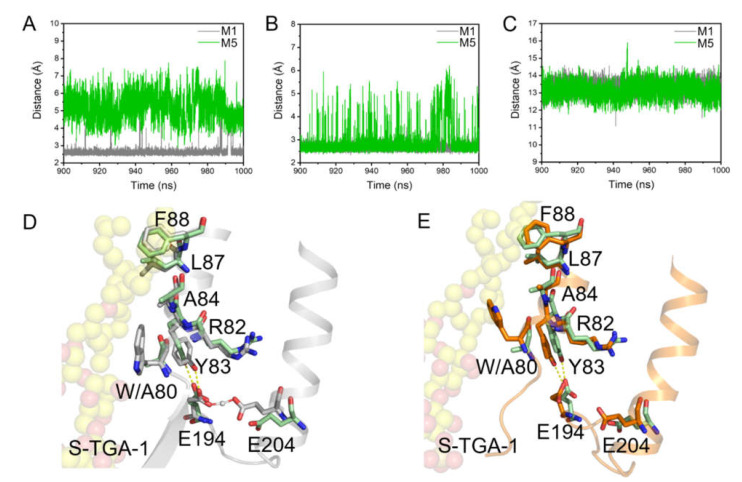
Dynamic conformation changes of the key residues on the extracellular side of bR induced by the W80A mutation. (**A**–**C**) The evolution of the distance between the key residues from the last 100 ns simulations of M1 (grey line) and M5 (green line): (**A**) E194-E204; (**B**) Y83-E194; (**C**) R82-A51. This behavior could be reproduced in the second time-repeat simulation (Appendix A and Appendix A); (**D**,**E**) superimposed drawing of several key residues on the extracellular side: (**D**) M1 (grey cartoon and sticks) vs. M5 (pale green sticks); (**E**) M2 (orange cartoon and sticks) vs. M5. Yellow spheres highlight S-TGA-1.

**Figure 8 ijms-23-06913-f008:**
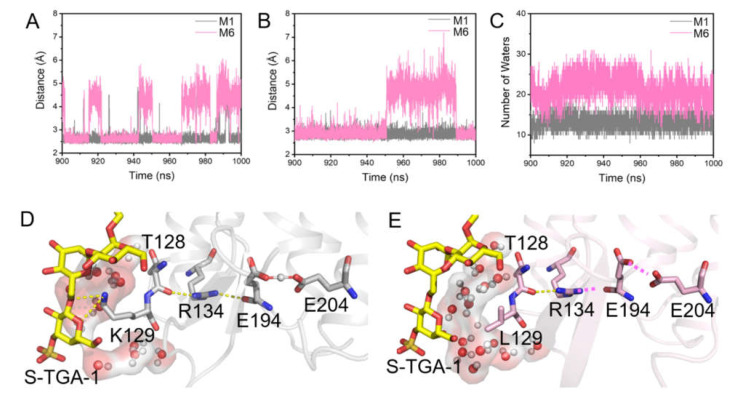
Dynamic conformational changes of the key residues on the extracellular side of bR induced by the K129L mutation. (**A**,**B**) The evolution of the distance between the key residues from the last 100 ns simulations of M1 (grey line) and M6 (pink line): (**A**) E194-E204; (**B**) R134-E194; (**C**) time series of the number of water molecules around K/L129 and T128 in the last 100 ns simulations of M1 (grey line) and M6 (pink line). This behavior could be reproduced in the second time-repeat simulation (Appendix A and Appendix A); (**D**,**E**) schematic diagrams of the H-bond network connecting K129 and the proton release complex (PRC) in M1 (**D**) and M6**(E)**. The grey cartoon and sticks were used for M1, pink cartoon and sticks were used for M6, yellow sticks represent S-TGA-1, red spheres, and the transparent surface indicates the water chains around K129 and T128, respectively. The pink dotted lines show unstable H-bond interactions.

**Figure 9 ijms-23-06913-f009:**
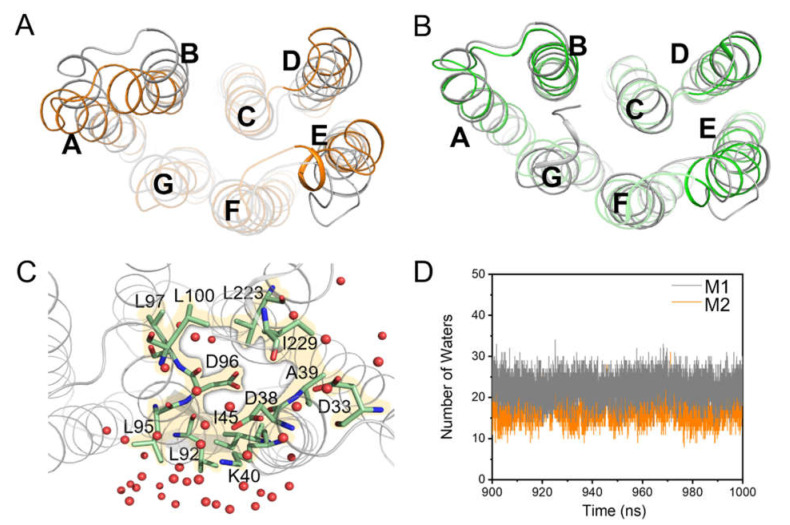
Dynamic changes in helices and water molecules on the cytoplasmic surface by the PGP-Me. (**A**) Superimposed drawing of the cytoplasmic side protein conformation of M1 (grey cartoon) and M2 (orange cartoon); (**B**) M1 and the late M state crystal structure of bR (PDB ID: 1FBB, green cartoon); (**C**) schematic diagram of the distribution of water molecules on the cytoplasmic surface from the last snapshot of M1. Red spheres represent the water molecules. D96 is surrounded by the key residues, especially hydrophobic ones (green stick and yellow surface). More water molecules gather around these residues, making it more accessible for them to enter the protein efficiently during the M state; (**D**) time series of the number of water molecules near the cytoplasmic surface in the last 100 ns simulations of M1 (grey line) and M2 (orange line).

**Figure 10 ijms-23-06913-f010:**
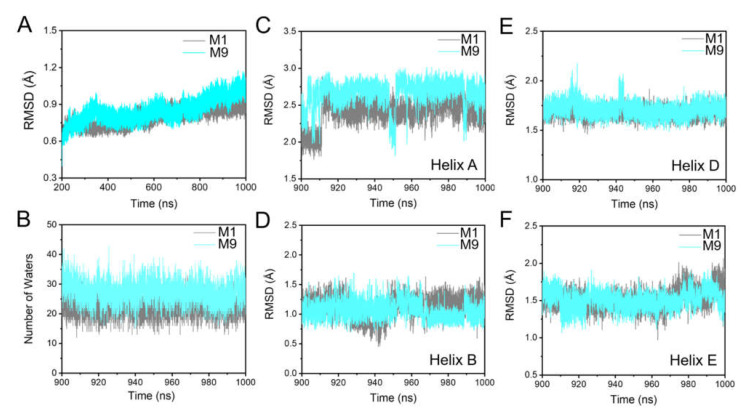
Dynamic changes in the helices and water molecules caused by the K40L mutation. (**A**) Time evolution of RMSD of the Cα atoms during the simulation of M1 and M9 (membrane of the K40L-bR trimer with POPC, S-TGA-1, and PGP-Me); (**B**) time series of the number of water molecules near the cytoplasmic surface in the last 100 ns simulations of M1 and M9; (**C**–**F**) the RMSD of the cytoplasmic end of helices A (**C**), B (**D**), D (**E**), and E (**F**) in M1 and M9 in a late M-state crystal structure (PDB ID: 1FBB). A smaller RMSD indicates a larger opening. Grey and cyan indicate the M1 and M9, respectively.

**Figure 11 ijms-23-06913-f011:**
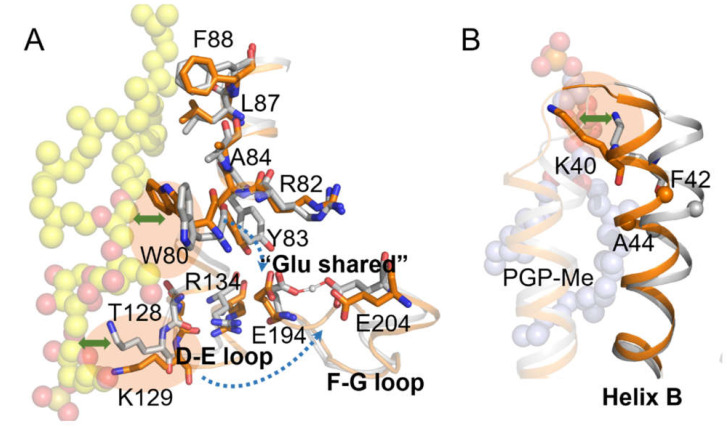
Schematic flowchart of the effects of S-TGA-1 (**A**) and PGP-Me (**B**) on the extracellular and cytoplasmic surfaces of bR. Yellow and light-blue spheres represent S-TGA-1 and PGP-Me, respectively. Grey and orange sticks indicate the regional conformation of bR and lipids in M1 and M2, respectively; green bidirectional arrows represent the specific interactions between residues and lipids, and blue dashed arrows indicate the regulatory role of W80 and K129 on the “Glu-shared” model.

## Data Availability

Data related to this paper may be requested from the authors.

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
