# Peer review of "Archaeal Lipids Regulating the Trimeric Structure Dynamics of Bacteriorhodopsin for Efficient Proton Release and Uptake"

_ijms, 2022, doi:10.3390/ijms23136913_

Round 1

Reviewer 1 Report

In this work Sijin Chen and coworkers study the function of two archaeal lipids, S-TGA-1 and PGP-Me, in the stabilization of the bR trimer by molecular dynamic simulations. They studied the interaction of these lipids with some of the key residues of bR, situated in its cytoplasmic and extracellular side. The quality of the data is sound, but the authors should clarify better some aspects of the work:

·         How many units of POPC and S-TGA-1 and PGP-Me were employed in the simulations and how many water molecules?

·         Table S1, the column named “System”, has errors for M1 and M2.

·         Page2, second paragraph: photocycle M2 instead of M2.

·         Page2, third paragraph: Is means is, right?

·         Figure 2: G, H and I, add label on the figure with the info M1 vs M2 for G, M3 vs M2 for H and M4 vs M2, to clarify and for the figure to be self-sufficient.

·         Figure 2: Error in the order of the letters, J and K are incorrectly placed.

·         Figure 5D: add amino acid A51 in the figure.

·         Maybe an error in the edition of the manuscript by the publisher but the caption in Figure 7 was the same size as the manuscript text.

·          Section 2.3 is a bit messy. First paragraph misses an introduction for the K40L mutant since the data displayed in Figure 8 are already on the K40L mutant.

·         Page 8, First paragraph in section 2.3, “helix” instead of helices

·         Caption Figure 8 A) is cytoplasmic no extracellular

§  Same for C)

·         Caption Figure 10: Write again the color coding even though it is the same as in Figure 5D

·         What is exactly Table S7? It’s not mentioned in the main manuscript.

Author Response

Reviewer #1:

Comments and Suggestions for Authors: In this work Sijin Chen and coworkers study the function of two archaeal lipids, S-TGA-1 and PGP-Me, in the stabilization of the bR trimer by molecular dynamic simulations. They studied the interaction of these lipids with some of the key residues of bR, situated in its cytoplasmic and extracellular side. The quality of the data is sound, but the authors should clarify better some aspects of the work:

How many units of POPC and S-TGA-1, and PGP-Me were employed in the simulations and how many water molecules?

Answer: We gratefully appreciate your comments and suggestions, which are very helpful in improving the quality of the manuscript.

The numbers of POPC and water molecules used in the simulations were provided in the original manuscript. We have added the corresponding information for S-TGA-1 and PGP-Me in the revised manuscript.

1) Table S1, the column named “System”, has errors for M1 and M2.

2) Page2, second paragraph: photocycle M2’ instead of M2.

3) Page2, third paragraph: Is means is, right?

4) Figure 2: G, H and I, add label on the figure with the info M1 vs M2 for G, M3 vs M2 for H and M4 vs M2, to clarify and for the figure to be self-sufficient.

5) Figure 2: Error in the order of the letters, J and K are incorrectly placed.

6) Figure 5D: add amino acid A51 in the figure.

7) Maybe an error in the edition of the manuscript by the publisher but the caption in Figure 7 was the same size as the manuscript text.

8) Page 8, First paragraph in section 2.3, “helix” instead of helices

9) Caption Figure 8 A) is cytoplasmic no extracellular § Same for C)

10) Caption Figure 10: Write again the color coding even though it is the same as in Figure 5D

11) What is exactly Table S7? It’s not mentioned in the main manuscript.

Answer: Many thanks for your careful reading and criticism. As shown in the revised manuscript, we have made all corrections accordingly.

Table S7 shows the forward and reverse primers of K40L-bop and K129L-bop.

Section 2.3 is a bit messy. First paragraph misses an introduction for the K40L mutant since the data displayed in Figure 8 are already on the K40L mutant.

Answer: Thank you very much for your comments and criticism. We have rephrased the first paragraph of Section 2.3 and rearranged the contents of several Figures accordingly. Please see the revised manuscript for all changes.

Reviewer 2 Report

Synopsis:

Lipid-protein interaction plays critical roles in regulating membrane protein functions and maintaining the quaternary structures of the functional units for membrane proteins. One of the current challenges to study lipid-protein interaction of bacteriorhodopsin (bR) is the limitation of archaeal lipid sources that are needed for protein function. Using a long-range molecular dynamics (MD) simulation, Chen et al in this study aimed to address the issue of how archaeal lipids maintain the trimeric conformation of bacteriorhodopsin in its native bacterial membranes, i.e., the purple membranes. The simulations predicted interactions of key residues on bR with two archaeal lipids, S-TGA-1 and PGP-Me, a result that will help generate hypotheses for future studies. However, some concerns require the attention of the authors, and this reviewer does not believe the manuscript in current format is convincing enough for its conclusion.

Concerns and suggestions:

1. The authors used POPC in the simulation, but this lipid is not native to the purple membranes. In fact, PG, instead of PC, is the primary source of phospholipids for the purple membranes. The authors failed to make a case of why their lipid bilayer system is relevant and why the results would reflect the near-true interaction in the purple membranes.

2. Despite the issue above, there are established and classic experimental approaches to evaluate the outcomes of this prediction, such as NMR or other biophysical methods using model membrane systems as used in the computer simulations.

3. The key question of this paper is to study how the trimeric bR is stabilized by S-TGA-1 and PGP-Me, but despite showing interaction between the lipids and the bR momomer, there was nearly no discussion about this subject. The authors did discuss a lot about proton release. In this reviewer’s humble view, the authors need to elaborate how the simulation results promote the trimeric conformation of bR in the membranes and whether such stabilization contributes to the later discussion about proton release.

4. The graphics are quite small to read. This reviewer needed to zoom in almost 300% on the computer screen to see the details, such as residue labeling.

Author Response

Reviewer #2:

Comments and Suggestions for Authors: Lipid-protein interaction plays critical roles in regulating membrane protein functions and maintaining the quaternary structures of the functional units for membrane proteins. One of the current challenges to study lipid-protein interaction of bacteriorhodopsin (bR) is the limitation of archaeal lipid sources that are needed for protein function. Using a long-range molecular dynamics (MD) simulation, Chen et al in this study aimed to address the issue of how archaeal lipids maintain the trimeric conformation of bacteriorhodopsin in its native bacterial membranes, i.e., the purple membranes. The simulations predicted interactions of key residues on bR with two archaeal lipids, S-TGA-1 and PGP-Me, a result that will help generate hypotheses for future studies. However, some concerns require the attention of the authors, and this reviewer does not believe the manuscript in current format is convincing enough for its conclusion.

We gratefully appreciate your comments, suggestions, and criticism, which are very helpful in improving the quality of the manuscript.

1) The authors used POPC in the simulation, but this lipid is not native to the purple membranes. In fact, PG, instead of PC, is the primary source of phospholipids for the purple membranes. The authors failed to make a case of why their lipid bilayer system is relevant and why the results would reflect the near-true interaction in the purple membranes.

Answer: Many thanks for your comments on this issue. To use of POPC in our simulations is based on the following considerations: 1) The topology and parameters of the POPC lipid bilayer model have been extensively tested [1]. 2) All the simulations of the bR trimer so far have been completed in the POPC lipid bilayer, and the results reflect the inter-residue interactions and water dynamics of the purple membrane (PM) well to a good approximation [2-6]. 3) X-ray and electron crystallography studies have clearly revealed the specific location and function of the archaeal lipids S-TGA-1 and PGP-Me in PM [7, 8]. 4) Neither POPC nor POPG can be assembled into the specific positions where the two archaeal lipids are located and fulfill their functions.

We agree that POPG may be better than POPC in providing a basic framework of the lipid bilayer for the simulations. However, in this paper, we explore the specific binding of S-TGA-1 and PGP-Me with the bR trimer and elucidate how these specific interactions modulate the bR trimeric structure and proton release and uptake. 

2) Despite the issue above, there are established and classic experimental approaches to evaluate the outcomes of this prediction, such as NMR or other biophysical methods using model membrane systems as used in the computer simulations.

Answer: Thank you for your comments. Studying the functions of S-TGA-1 and PGP-Me by biophysical methods requires comparing the PM mimic with and without the two archaeal lipids. However, due to the difficulty and high cost of synthesizing those two lipids, it is hard to obtain them commercially or preparing in the laboratory. So far, only one group has managed to isolate them from PM [9, 10]. On top of that, studying the interaction of S-TGA-1 and PGP-Me with bR by NMR requires 13C labeled the two archaeal lipids, making it even more challenging and costly to access. So, MD simulation is at least a good choice to explore the role of such specific archaeal lipids realistically.

3) The key question of this paper is to study how the trimeric bR is stabilized by S-TGA-1 and PGP-Me, but despite showing interaction between the lipids and the bR monomer, there was nearly no discussion about this subject. The authors did discuss a lot about proton release. In this reviewer’s humble view, the authors need to elaborate how the simulation results promote the trimeric conformation of bR in the membranes and whether such stabilization contributes to the later discussion about proton release.

Answer: Thank you for your comments. Actually, the individual interaction of S-TGA-1 and PGP-Me with a single bR monomer does not exist since S-TGA-1 locates in the center of the bR trimer closure and is essential for intra-trimer stabilization, PGP-Me exists in the space of the adjacent bR monomers acting as a glue to bridge contact between the monomers to maintain the trimer stabilization (Figure 1). All our simulations in this paper were performed on the bR trimer with different lipid mixtures (M1 to M10).  

In Section 2.1, we first discussed the effects of S-TGA-1 and PGP-Me on the overall stability and conformational dynamics of the bR trimer. Then in Sections 2.2 and 2.3, we explicitly explored the molecular mechanism of how these dynamic conformational changes induced by the specific binding of S-TGA-1 and PGP-Me reflected on the extracellular and cytoplasmic surfaces and affected the proton release and uptake through interacting with the particular residues.

4) The graphics are quite small to read. This reviewer needed to zoom in almost 300% on the computer screen to see the details, such as residue labeling.

Answer: Thank you for your criticism, and we apologize for that. We have enlarged and adjusted all the original graphics and related labels in the main text and supplementary material in the revised manuscript.

We appreciate all comments and have rephrased and added some sentences to enhance the statements in the revised manuscript.

References

  1. Feller, S.E.; MacKerell, A.D. An Improved Empirical Potential Energy Function for Molecular Simulations of Phospholipids. J. Phys. Chem. B. 2000, 104, 7510-7515.
  2. Kandt, C.; Gerwert, K.; Schlitter, J. Water dynamics simulation as a tool for probing proton transfer pathways in a heptahelical membrane protein. Proteins: Struct. Funct. Bioinform. 2005, 58, 528-537.
  3. Kandt, C.; Schlitter, J.; Gerwert, K. Dynamics of Water Molecules in the Bacteriorhodopsin Trimer in Explicit Lipid/Water Environment. Biophys. J. 2004, 86, 705-717.
  4. Grudinin, S.; Büldt, G.; Gordeliy, V.; Baumgaertner, A. Water Molecules and Hydrogen-Bonded Networks in Bacteriorhodopsin—Molecular Dynamics Simulations of the Ground State and the M-Intermediate. Biophys. J. 2005, 88, 3252-3261.
  5. del Val, C.; Bondar, L.; Bondar, A.-N. Coupling between inter-helical hydrogen bonding and water dynamics in a proton transporter. J. Struct. Biol. 2014, 186, 95-111.
  6. Freier, E.; Wolf, S.; Gerwert, K. Proton transfer via a transient linear water-molecule chain in a membrane protein. Proc. Natl. Acad. Sci. U.S.A. 2011, 108, 11435-11439.
  7. Essen, L.; Siegert, R.; Lehmann, W.D.; Oesterhelt, D. Lipid patches in membrane protein oligomers: crystal structure of the bacteriorhodopsin-lipid complex. Proc. Natl. Acad. Sci. U.S.A. 1998, 95, 11673-11678.
  8. Mitsuoka, K.; Hirai, T.; Murata, K.; Miyazawa, A.; Kidera, A.; Kimura, Y.; Fujiyoshi, Y. The structure of bacteriorhodopsin at 3.0 Å resolution based on electron crystallography: implication of the charge distribution. J. Mol. Biol. 1999, 286, 861-882.
  9. Inada, M.; Kinoshita, M.; Matsumori, N. Archaeal Glycolipid S-TGA-1 Is Crucial for Trimer Formation and Photocycle Activity of Bacteriorhodopsin. ACS Chem. Biol. 2020, 15, 197-204.
  10. Inada, M.; Kinoshita, M.; Sumino, A.; Oiki, S.; Matsumori, N. A concise method for quantitative analysis of interactions between lipids and membrane proteins. Anal. Chim. Acta. 2019, 1059, 103-112.

Round 2

Reviewer 2 Report

The authors have addressed all concerns as raised by this reviewer previously. It remains unclear, though, how this study may be broadly applied to other biological or physiological questions. It may be worthwhile to include such an outlook when finalizing the manuscript, as this is for a special issue in the advancement of molecular dynamics.

Author Response

Reviewer #2:

Comments and Suggestions for Authors: The authors have addressed all concerns as raised by this reviewer previously. It remains unclear, though, how this study may be broadly applied to other biological or physiological questions. It may be worthwhile to include such an outlook when finalizing the manuscript, as this is for a special issue in the advancement of molecular dynamics.

We appreciate your kind suggestion and have added a short paragraph at the end of the conclusion part as follows:

This work demonstrates that the long-term and all-atom molecular dynamics simulation is a powerful method to connect the overall structural stability and conformational dynamics of a large membrane protein complex with the dynamic function of the protein. The strategy used here may be broadly applied to study the dynamic regulation of the function-related conformation changes induced by the specific binding of other membrane receptors, such as lipids or ligand bindings.
